# Closing Data Gaps to Measure the Bioeconomy in the EU

**Vineta Tetere [1,2,*] and Sandija Zeverte-Rivza [2]**

1  Agricultural Economics and Rural Policy Group, Wageningen University & Research, Hollandseweg 1, 6706 KN Wageningen, The Netherlands
2  Faculty of Economics and Social Development, Latvia University of Life Sciences and Technologies, Svetes Str 18, LV-3001 Jelgava, Latvia
*  Correspondence: vineta.tetere@wur.nl; Tel.: +371-26543783

**Abstract:** The expansion of bio-based value chains is prioritized through various European Union (EU) policy initiatives. Due to the growing awareness of the importance of a sustainable bioeconomy in Europe, the need to increase the availability and quality of statistics is increasing. There are several essential aspects lacking, including (i) comprehensive databases and statistics for bio-based sectors; (ii) transparent methodology for bio-based data collection; and (iii) integrated value chain data and indicators that illustrate the flows of different bio-based commodities. The aim of this paper is to develop a bio-based material flow monitor to measure the physical contribution of industries to the bioeconomy. The material flow monitor describes physical material flows (including biomass) to, from, and within the economy. It is recorded in the form of supply and use tables. To measure the bioeconomy, the BioSAM database, along with disaggregated commodities and activities, are used. Data regarding waste generation/treatment and $CO_2$ emissions/sequestrations are added to assess the impact on climate change. The results indicate that the bioeconomy in the EU is underreported due to a lack of data, leading to an insufficient understanding of its contribution to the economy. It can also be concluded that the data from the BioSAM tables are the most complete and have the highest disaggregation level for commodities and sectors, allowing one to measure the significance of the bioeconomy.

**Keywords:** commodity; sector; data; biomass; bioeconomy





## 1. Introduction

The bioeconomy comprises different sectors of the economy that produce, process, and re-use renewable biological resources. These sectors are typically agriculture, forestry, fishing, chemicals, food, bio-based materials, and bioenergy. Therefore, it is in our interest to analyze the possible impacts of sectoral policies at the national and regional levels, as well as cross-sectoral policies that address social challenges such as increasing food demand and climate change [1]. The European Commission defines the scope of bioeconomy as encompassing all sectors and systems that rely on biological resources (animals, plants, micro-organisms, and derived biomass, including organic waste), their functions, and their principles. Interlinks are also included, such as land and marine ecosystems and the services they provide; all primary production sectors that use and produce biological resources, i.e., agriculture, forestry, fisheries, and aquaculture; and all economic and industrial sectors that use biological resources and processes to produce food, feed, bio-based products, energy, and services [2].

A major barrier to analyzing the activities of the bioeconomy is the lack of available data. More specifically, in the framework of the standard National Accounts, bioeconomy sectors are represented as broad sectoral aggregates (i.e., agriculture, food processing, forestry, fisheries, wood, and pulp) or even subsumed within their parent sectors (for example, chemical industries, clothing, energy). Furthermore, while the coverage of secondary data on traditional bioeconomy activities is 'relatively' accessible, data inventories

of more contemporary biomass sources and bio-based activities are scarce, which severely hampers our ability to estimate the economic importance of these evolving sectors within national economies [3].

While bio-based products and materials are on the rise, it is still difficult to quantify and monitor the development of the bioeconomy. This is because bio-based materials and products are increasingly used to substitute petrochemicals, but are not separately captured by statistical classifications. Using coefficients to determine the bio-based shares per product category is the best method currently available to measure the size of the bioeconomy, but it is not sufficiently accurate to reliably capture small developments [4].

Numerous attempts have been made to measure the size of the bioeconomy. The most widely used indicators to monitor the impact of national bioeconomy strategies and to measure the size of bioeconomy include the gross output (turnover), value added, investments, exports of bioeconomy goods, and employment [5]. Cingiz et.al. [6] analyzed the value added of the bioeconomy in 28 EU member states using an input–output (IO) model. The IO model is suitable for tracking biomass inputs, for determining the contribution of different industries to the bioeconomy, and, consequently, for assessing the bioeconomy's contribution to the total economy. Lazorcakova et.al. [7] used IO analysis to quantify economic as well as environmental indicators for the purpose of measuring the bioeconomy. These studies involve a high aggregation level of commodities and sectors that do not display physical volumes of specific bio-based commodities.

The Joint Research Centre (JRC) of the European Commission (EC) developed the biomass flow diagram that describes the flow of biomass across sectors of the economy [8]. In this case, the physical flows were directly estimated without conversion from monetary data at a higher level of sectorial aggregation. Meanwhile, the Material Flow Monitor (MFM) was developed by Statistics Netherlands [9], and describes the physical material flows of the Dutch economy, measured in millions of kilograms. Physical flows also include imports and exports of goods. The MFM originates from the monetary supply and use tables published in the National Accounts, and is converted into physical flows of materials. The level of disaggregation achieved with these material flows makes it possible to estimate the size of the bioeconomy, in physical terms, relative to the entire Dutch economy. Currently, this tool is only available for the Netherlands.

The aim of this paper is to develop a bio-based material flow monitor (BFM) to measure the physical contributions of different sectors to the bioeconomy of Latvia. It is structured as follows. Section 2 describes the bioeconomy in the EU and Latvia. Section 3 presents the method and data. Section 4 presents and discusses the results obtained using the method. Section 5 concludes the paper and provides a general discussion and limitations.

## 2. Measuring Bioeconomy in EU and Latvia

The bioeconomy is an important part of the EU economy. In 2018, the total number of employed persons in the EU bioeconomy amounted to 18.4 million, with a declining trend which was mainly due to the decrease in employment in the agricultural sector, caused by consolidation across the agricultural sector [10] and increasing efficiency through optimization, automation, and digitalization of the sector [11,12]. The data for EU-28 show a continuous increase in turnover in the bioeconomy from less than EUR 2 trillion in 2008 to more than EUR 2.43 trillion in 2018, with the food sector being the largest contributor [13]. The focus on the potential of the bioeconomy highlighted in the EU policy narratives (e.g., EU Bioeconomy strategy and Green Deal) makes it essential to monitor the bioeconomy and to understand its driving forces.

For Latvia, bio-based sectors are an essential part of the national economy. In 2018, the Ministry of Agriculture of Latvia defined the Latvian Bioeconomy Strategy for 2030 [14], with an aim to become the EU leader in the preservation, growth, and sustainable use of natural capital. The Strategy also recognizes the necessity of raising awareness of the significance of the bioeconomy in Latvia and of determining the potential directions for development up to 2030. Thus, it also indicates the need for a comprehensive data collection

and analysis of the bioeconomy that would enable researchers to monitor the bioeconomy sectors more thoroughly.

According to the Strategy, the bioeconomy includes agriculture, forestry, fisheries, aquaculture, production of food, cellulose, and paper, as well as, in part, the chemical, biotechnology, and energy sectors. This definition excludes other industries that use bio-based inputs in production. The objectives of this strategy are to maintain or increase exports and value provided by traditional bioeconomy sectors, namely agriculture (A01); forestry (A02); fisheries (A03); processing of food, beverages, and tobacco (C10, C11, C12); manufacturing of leather and related products (C15); manufacturing of wood, straw, plaiting materials, and paper products (C16, C17); and manufacturing of furniture (C31). Using the structure of the bioeconomy defined in the Strategy is the only approach to measure the size of the bioeconomy of Latvia. Furthermore, it is expressed in monetary values, and volatility of price can create biased increases in these values. Therefore, it would be advisable, rather, to use physical volumes, as they would provide an unbiased outlook on the size of the bioeconomy.

In Table 1, the main indicators of the bioeconomy are shown for EU-27 and Latvia.

**Table 1.** Employment, gross value added, and turnover per person employed in the bioeconomy in EU-27 and Latvia in 2015 and 2019 [15].

| Indicator | EU | | LV | |
|---|---|---|---|---|
| | **2015** | **2019** | **2015** | **2019** |
| Employment, 1000 persons | 17,691.34 | 17,424.29 | 132.06 | 120.46 |
| GVA per employed person, EUR 1000/person | 31.40 | 37.70 | 14.80 | 20.80 |
| Turnover per employed person, EUR 1000 | 116.91 | 134.62 | 52.31 | 66.72 |

Table 1 shows that the size of the bioeconomy is growing, both in EU and in Latvia. Gross value added (GVA) and turnover per employed person are higher in the EU than in Latvia, while the bio-based share of the bioeconomy is above 80% in Latvia and above 60% in the EU.

In both the EU and Latvia, the largest share of value added is created by traditional bioeconomy sectors such as agriculture and food production, but other sectors also contribute (Figure 1). In Latvia, a large share of the bioeconomy (48.7%) is taken up by forestry and production of wood products and furniture. This share is considerably higher than the EU average in these sectors. It is also noticeable that more advanced sectors with a higher value added (e.g., bio-based pharmaceuticals and bio-based chemicals) are not as prominent in the bioeconomic structure of Latvia compared to the EU average. These tendencies indicate a lower value added for the produced goods within the national bioeconomy, and are also visible in the data regarding the GVA and turnover per employed person, as is reflected in Table 1.

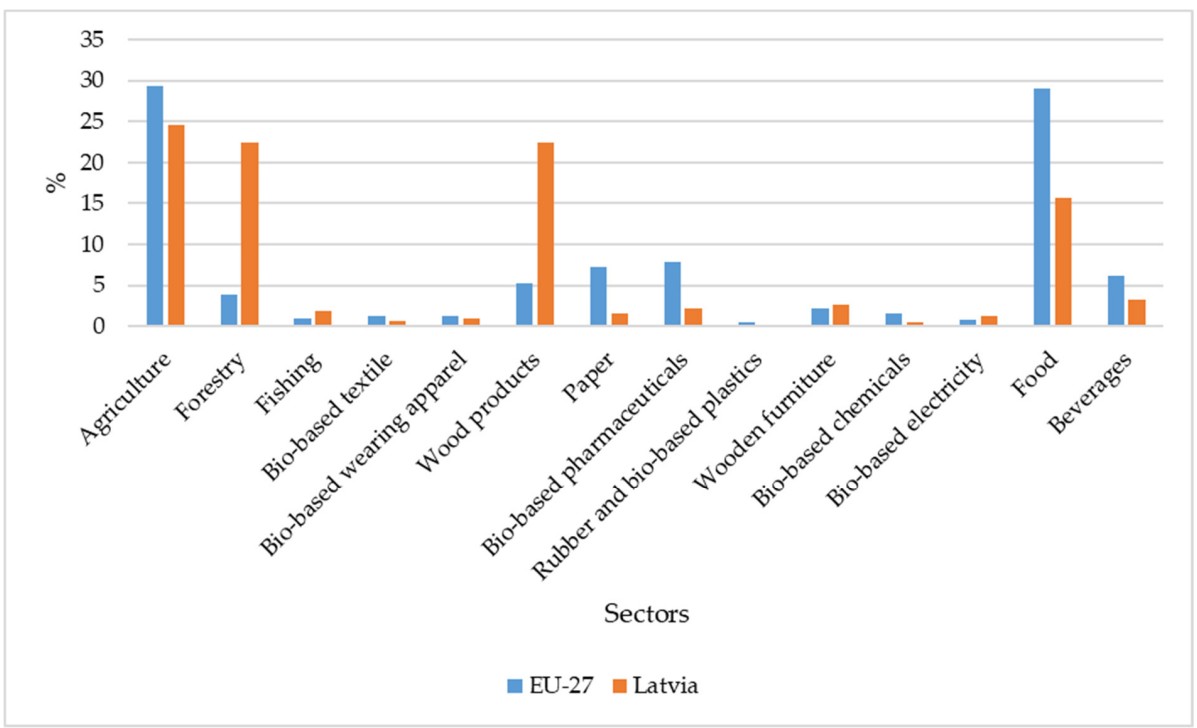

**Figure 1.** Value added per sector in the bioeconomies of EU-27 and Latvia in 2019, in % [11].

## 3. Materials and Methods

### 3.1. Data

According to the definition of the European Commission [2], the bioeconomy encompasses not only the primary sectors that employ and produce biological resources, but also the activities that depend on biological resources to generate value added products (i.e., food, feed, materials, and energy).

Crucial data need to be gathered regarding the methodology of compiling the bio-based material flow monitor (BFM) to measure the significance of the bioeconomy. Detailed monetary supply and use tables (MSUT), as well as unit values, are the basic data required. There are several options for dealing with unavailable MSUT. The first option is to take the 64 commodities, according to the SUT of 64 sectors, that every national statistics institution reports to Eurostat. An advantage of this is the availability of data, because the data are published annually by Eurostat, while a disadvantage is that the aggregated table gives very rough estimates when converting to physical units. The reason for this is that the commodities and sectors are very heterogeneous, even within a single group. Therefore, it is difficult to find a unit value that represents a group. The 64-by-64 tables can further be disaggregated by using the PRODCOM dataset. An advantage of this is that it is a detailed dataset, but disaggregation of only the supply side is possible, as there are no data for the use table. PRODCOM dataset has limited (confidential) data in the publicly available statistical files, especially for a small-sized country like Latvia. Therefore, PRODCOM dataset was not used in this research. The Exiobase dataset is another option for obtaining detailed supply and use tables of a country. However, the Exiobase has not been updated for a long time and is, therefore, not up-to-date.

Another option is to use a BioSAM (Social Accounting Matrix) developed by EC JRC [16]. The advantage is that it is publicly available, while the disadvantage is that the use and supply should be separated, and there is no direct link with the monetary data of the National Accounts [9]. For each of the EU member states, and for the EU aggregate, a set of highly disaggregated bio-based sector account splits within the framework of BioSAM were constructed for the first time for the year 2010. Since then, BioSAM has been constructed every five years, with the latest version developed in 2015. The systematic

method of estimation is used based on the reconciliation of four main databases, namely, (i) the complete and consistent (CoCo) database from the Common Agricultural Policy Regionalized Impacts (CAPRI) model, using re-estimated AgroSAMs (Table A1); (ii) the National Accounts; (iii) the Economic Accounts for Agriculture (EAA) from Eurostat; and, finally, (iv) the database of the computable general equilibrium model (MAGNET) [1].

In addition, diverse types of data sources were used to compile unit values (prices) for domestically produced commodities. First, sectoral data from the national statistics [17], and second, data on international trade [18], were used to develop unit values for domestically produced and used commodities. Unit values of international trade statistics are a quantity-weighted average of the different prices at which the product is purchased/imported or sold/exported. Unit values of imported goods are given in the use table, and unit values of exported goods are given in the supply table.

Data on $CO_2$ emissions [19] and waste generation [20] and treatment [21] were included from Eurostat. Due to biannual data reports, waste management data were taken from 2016.

To calculate bio-based shares of the supplied/used commodities, a variety of data sources, e.g., nova-Institute, Statistics Netherlands, Classification of Products by Activity (CPA), and 4-digit aggregation of commodities (where available) were used based on international trade data and unpublished researchers' opinions (Table A3).

### 3.2. Methods

This article reflects an attempt to apply the approach developed by Statistics Netherlands (SN) to the datasets available for Latvia, intending to become one of the pilot countries to test the validity of this approach. For this analysis, BioSAM data from 2015 were used. To the best of our knowledge, for the EU and its member states, including for Latvia, this dataset is the most complete multisector database that exists with detailed coverage on the bioeconomy sectors and their links with other activities and institutional sectors.

The approach of SN was combined with the BioSAM dataset and price information from different sources to measure the bio-based material flow in Latvia. SN instead used the SUT and price information with a high disaggregation level which they collected for the Netherlands.

After separating the supply and use of BioSAM, a structure was obtained for commodities, services, and sectors, and is reflected in Tables 2 and 3.

**Table 2.** Structure of the supply table.

| Supply, Million EUR | Sector 1 | Sector 2 | Imports | Total |
|---|---|---|---|---|
| Commodity 1 | 2 | 5 | 5 | 12 |
| Commodity 2 | 6 | 3 | 2 | 11 |
| Service 1 | 1 | 0 | 0 | 1 |
| Total | 9 | 8 | 7 | 24 |

**Table 3.** Structure of the use table.

| Use, Million EUR | Sector 1 | Sector 2 | Exports | Total |
|---|---|---|---|---|
| Commodity 1 | 7 | 4 | 0 | 11 |
| Commodity 2 | 2 | 3 | 7 | 12 |
| Service 1 | 0 | 3 | 0 | 3 |
| Total | 9 | 10 | 7 | 26 |

Initially, BioSAM contained 171 accounts for 2015, including 80 activity/commodity accounts (see Table A2). According to the available data for Latvia, there were 63 commodities and 69 activities/sectors and households. In the supply table, an additional column for other activities (Table A4) was added into which allocate supplied volumes which were not

mentioned elsewhere in BioSAM accounts. In addition, the BioSAM contained one account for the rest of the world (exports/imports).

As a next step, unit values were used to calculate physical supply and use tables (PSUT) using monetary supply and use tables (MSUT) as a basis. The unit value of a commodity is assessed in EUR per kilogram (kg). For this reason, services are excluded from the PSUT for Latvia.

To calculate physical (*p*) supply (*S*) of a commodity (*i*) by sector (*j*) in millions of kg, the unit price (*p*) of the commodity was applied to the monetary (*m*) supply of the commodity supplied by the sector, in increments of EUR 1 million.

$$S_{ij}^{p} = \frac{S_{ij}^{m}}{P_i} \tag{1}$$

For the calculation of the use (*U*) in millions of kg, the same approach was used.

$$U_{ij}^{p} = \frac{U_{ij}^{m}}{P_i} \tag{2}$$

The PSUT can be enhanced by adding additional information (rows and columns). Conversion from MSUT to PSUT has not yet provided a complete overview of all the material flows. Material flows may exist where there is no monetary component in the MSUT, such as $CO_2$ emissions or waste. These flows are include in the MFM and are used for balancing the sectors.

Some production processes use resources that are extracted from the environment. For example, trees for the timber industry or grain harvests in agriculture are also considered extraction. Data on extraction were taken from Material Flow Accounts (MFA). The MFA covers the extraction of crops, livestock crops, wood, fish, salt, limestone, clay, sand, gravel, natural gas, and crude oil.

Due to differences in sources and in the quality of the data, it was possible that supply would not equal use, even with balancing items (BI) such as emissions and waste. Therefore, the last step to complete the MFM was to balance the supply and use of goods and the input and output of sectors. Supply was expected to equal use for each product group (rows), because logically, all materials supplied in the economy must be used. Likewise, for each sector (columns), the amount of materials used was also supplied in one form or another. The result of this methodology was a complete and balanced physical supply and use table (see Tables 4 and 5).

**Table 4.** Example of a balanced supply table.

| Supply, million kg | Sector 1 | Sector 2 | Imports | Environment | BI | Total |
|---|---|---|---|---|---|---|
| Commodity 1 | 2 | 5 | 5 | | 0 | 12 |
| Commodity 2 | 6 | 3 | 2 | | 1 | 12 |
| Natural extractions | | | | 10 | 0 | 10 |
| $CO_2$ emissions | 2 | 1 | 0 | 0 | 5 | 8 |
| Waste generation | 4 | 6 | 2 | 0 | 0 | 12 |
| Balancing item | 1 | 0 | 0 | 1 | 0 | 2 |
| Total | 15 | 15 | 9 | 11 | 6 | 56 |

The background is made to show that those lines should not contain any data.

After balancing the tables, the bio-based shares (Table A3) were applied to commodities to obtain physical bio-based material flow monitor data (BFM). Using the waste generation dataset, part of the waste was indicated as bio-based (paper and cardboard wastes; wood wastes; textile wastes; animal and mixed food waste; and animal feces, urine, and manure). The shares of these waste categories were used to calculate bio-based waste. Calculating the share of bio-based energy carriers (biodiesel, biochemicals, and biogas) provides the

option to calculate bio-based $CO_2$ emissions. The coefficients were fixed in this study, but they may vary by sector in practice.

**Table 5.** Example of a balanced use table.

| Use, million kg | Sector 1 | Sector 2 | Exports | Environment | BI | Total |
|---|---|---|---|---|---|---|
| Commodity 1 | 7 | 4 | 0 | | 1 | 12 |
| Commodity 2 | 2 | 3 | 7 | | 0 | 12 |
| Natural extractions | 6 | 4 | | | 0 | 10 |
| $CO_2$ sequestration | | | | 8 | 0 | 8 |
| Waste treatment | 0 | 3 | 1 | 3 | 5 | 7 |
| Balancing item | 0 | 1 | 1 | 0 | 0 | 2 |
| Total | 15 | 15 | 9 | 11 | 6 | 56 |

The background is made to show that those lines should not contain any data.

Figure 2 presents a summary of the method applied in this research.

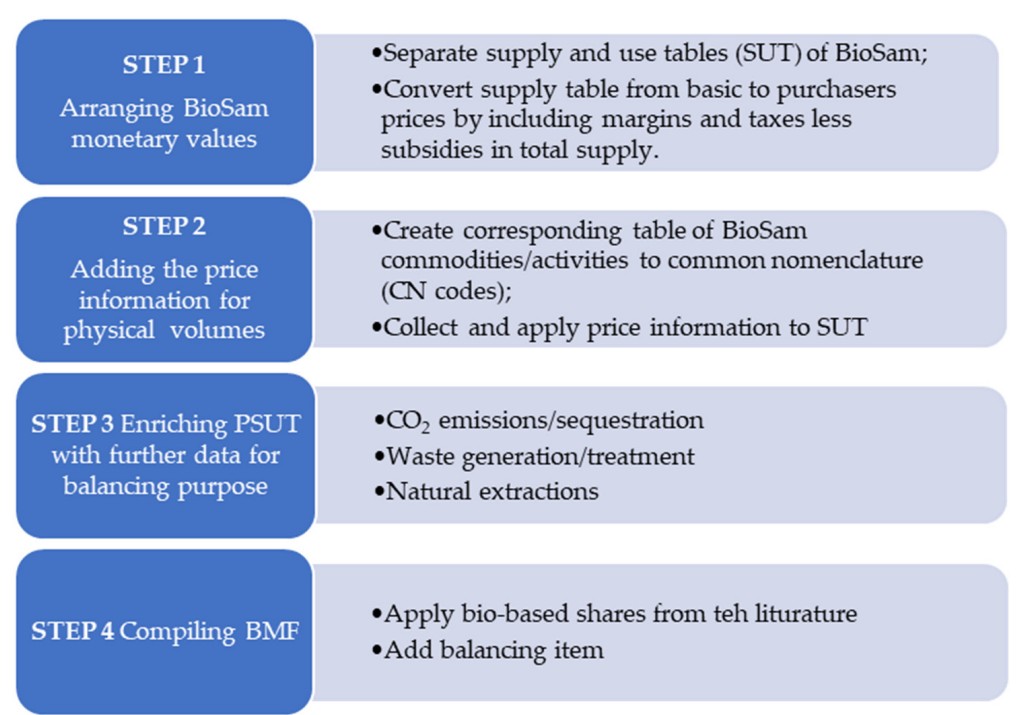

**Figure 2.** Steps to convert the BioSAM dataset from MSUT to PSUT for BFM.

## 4. Results and Discussion

The application of the methodology previously described for the data on the Latvian bioeconomy enabled us to review the contribution of bio-based material flows to the bioeconomy and to distinguish the bio-based and non-bio-based material flows within the sectors.

There are sectors such as construction, administration, education, and manufacturing that are not considered in Knowledge Center data nor in the Latvian Bioeconomy Strategy, but these sectors are included in the BSUT of BioSAM.

Few assumptions have been made to estimate the results. First, from a commodity perspective, a value pyramid based on the financial value of biomass applications developed by Bos et al. [22] (Figure 3) was used. Each commodity is allocated to a category irrespective of which sector uses the commodity, and all waste is allocated to residual flows without considering if it is used for energy generation, fodder, or materials.

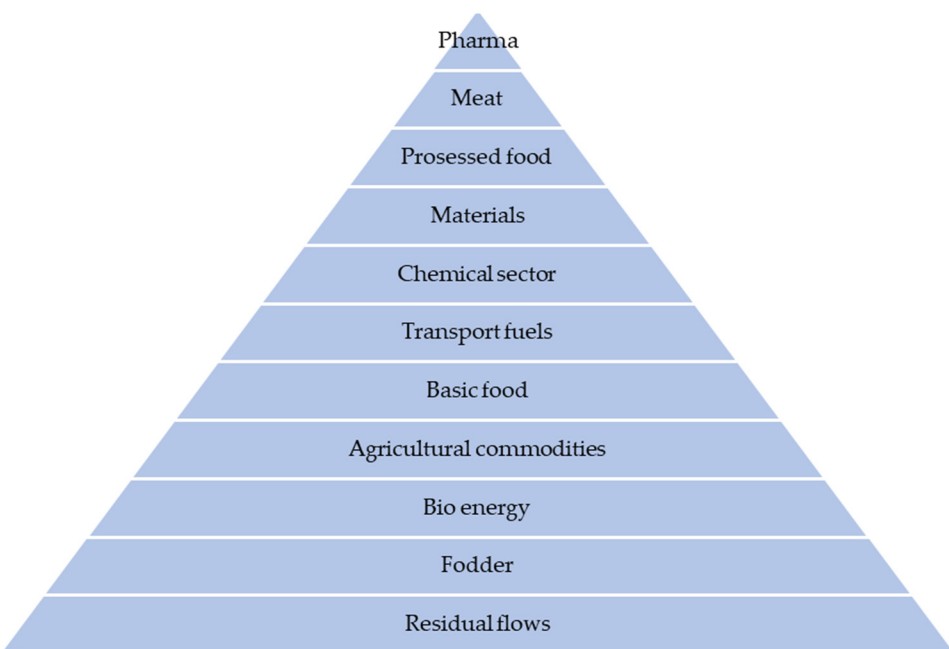

**Figure 3.** Value pyramid adapted from Bos et al. [22].

The layer at the bottom of the pyramid represents large volumes of relatively low-value biomass. Higher in the pyramid, biomass that has been transformed into other products is represented; these types typically have smaller volumes, but higher values.

When commodities are cascaded in Latvia, according to the value hierarchy developed by Bos et al. [22] (Figure 4), it can be observed that materials form the share with the largest value (EUR 48.78 billion), followed by processed food (EUR 41.77 billion) and agricultural commodities (EUR 34.50 billion). The lowest value is formed by the production of transport fuels (EUR 0.34 billion).

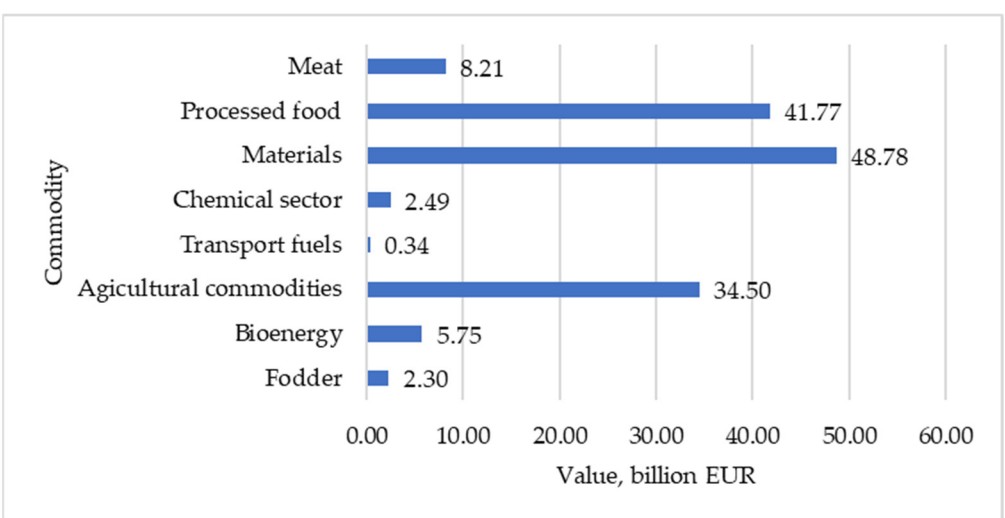

**Figure 4.** Cascading of biomass in Latvia in 2015, in EUR 1 billion.

Agriculture commodities form by far the largest share of volume (436.22 billion kg). These commodities are mainly used by the agriculture sector itself (40%) and the food processing sector (20%). More advanced materials, such as transport fuels and chemistry-related bio-based commodities, form the smallest share (0.49 and 0.91 billion kg, respectively). In general, most biomass is contained in the bottom half of the pyramid. Nevertheless, the category "Materials", in the top half of the pyramid, forms a relatively large

share as well. As the wood processing sector is prevalent in the bioeconomy of Latvia, commodities in this category, namely, paper, wood products, and textiles, highlight the specialization of the bioeconomy sector of Latvia in wood processing.

The cascade of biomass in Latvia includes all types of commodities for which biomass is used, except pharmaceuticals and basic food, which are not included due to the aggregation level of the BioSAM data. It can be observed that agriculture commodities comprise the largest share in the cascade of biomass, followed by materials, which is not in accordance with the value pyramid by Bos et al. [22]; however, it is expected for Latvia due to the high availability and use of forest biomass.

The sectors compared in Figure 5 from the BioSAM dataset form 87.8% of the total bioeconomy, and when compared to the data from the knowledge center, it can be observed that there are several sectors, e.g., forestry, where differences are as large as almost 10%.

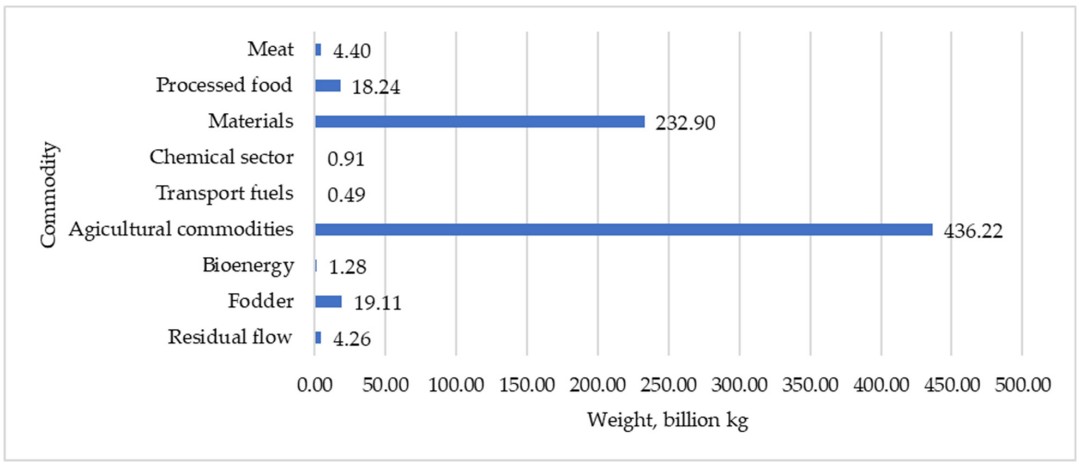

**Figure 5.** Cascade of biomass in Latvia in 2015, in billion kg.

The results indicate that there are more sectors contributing to the bioeconomy than reported. Figure 6 demonstrate the results of the study, which show that the share of the bioeconomy's sectors differ according to the data source.

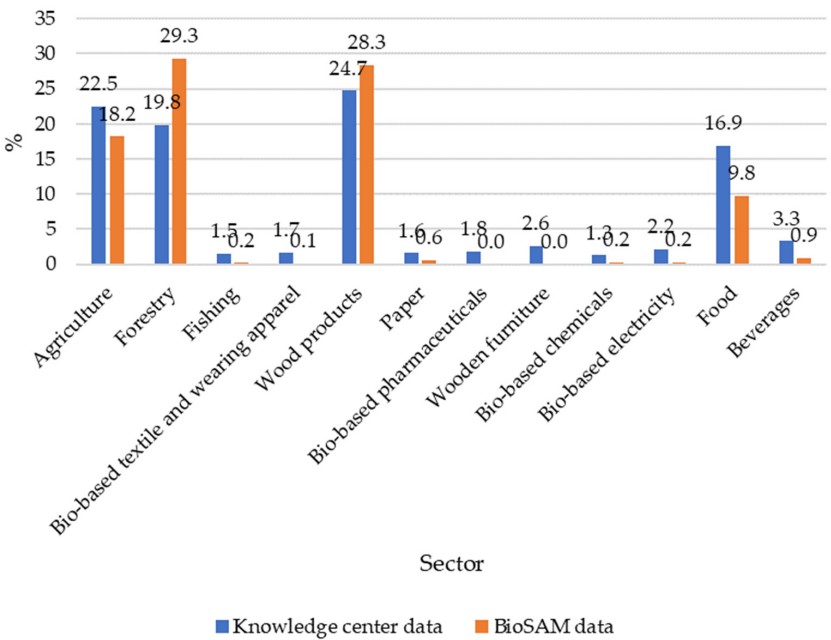

**Figure 6.** Share of sectors in the bioeconomy in Latvia, comparison of results of PSUT of BioSAM and Knowledge Centre data in 2015, in % [15].

Furthermore, the results of the adopted methodology can be used to describe the material flows of Latvia's economy. Figure 7 reflects the material flow between the supply and use of bio-based and non-bio-based commodities, carbon dioxide ($CO_2$) emissions, and waste. In total, 49.79% of all commodities are domestically supplied and 50.21% are imported; of that proportion, 93.18% non-bio-based commodities are imported. Latvia's bioeconomy is largely self-sufficient in terms of the use of bio-based commodities, with only a minor flow of bio-based commodity imports (6.81% of all imports).

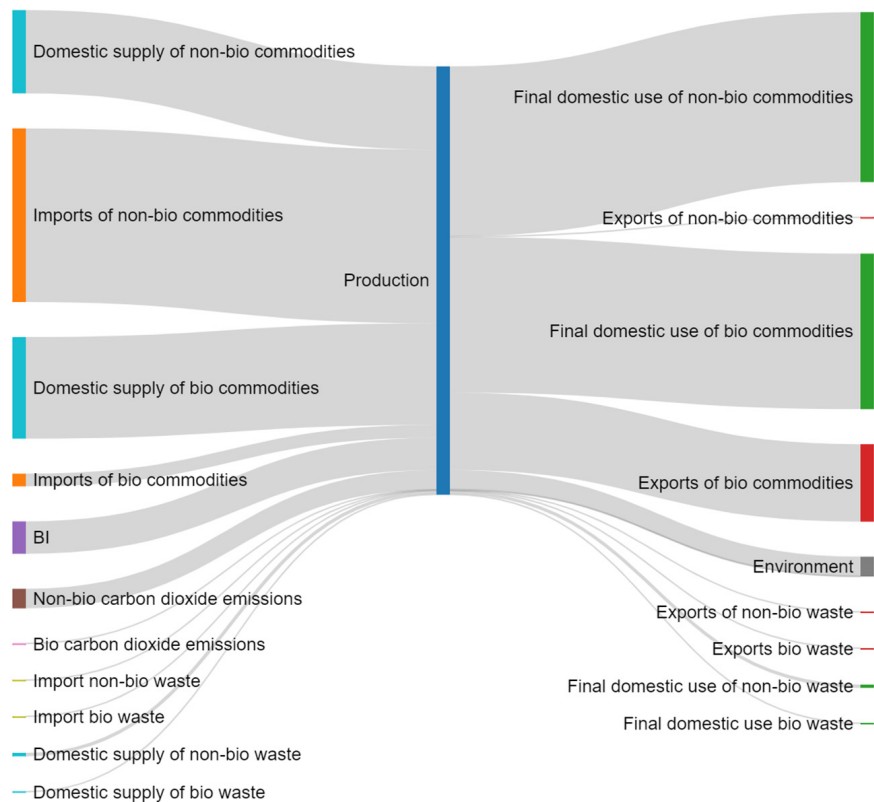

**Figure 7.** Bio-based and non-bio-based material flow (Sankey diagram) of the economy in Latvia in 2015, in million kg.

On the use side, it can be observed that the produced commodities are mainly used domestically—80.66%—while 19.34% are exported. Of those exported, 99.19% are bio-based commodities, and only 0.8% are non-bio-based commodities.

Material flows can be also constructed for each sector of the economy or each commodity separately (e.g., "Fertilizers" in Figure 8). In this case, the material flow shows the supply and use of bio-based and non-bio-based streams of fertilizers within the economy.

Figure 8 indicates that the material flow of fertilizers is dominated by the import of non-bio-based fertilizers (167.83 million kg, or 99.9% of all imports and 68.67% of all supply). The import of bio-based fertilizers is only 0.17 million kg, or 0.1% of all imports of fertilizers. Domestic supply is also dominated by non-bio-based supply—76.29 million kg or 99.9% of the domestic supply—and only 0.08 million kg of bio-based fertilizers are supplied domestically. Exports form 32.81% of the total use, and 99.3% of those are non-bio-based, while 0.07% are bio-based fertilizer exports. In addition, 61.29% of all fertilizers are used domestically, with the same proportion of bio-based and non-bio-based fertilizers as are exported. A total of 14.41 million kg or 5.9% of fertilizers are stored domestically as stock.

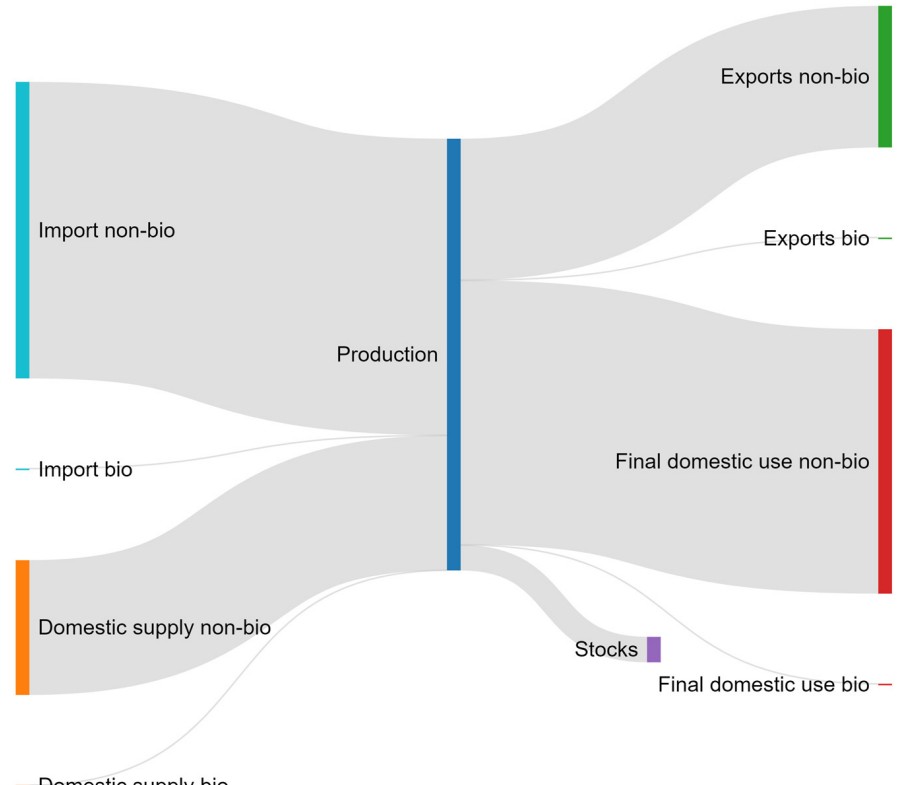

**Figure 8.** Material flow of bio-based and non-bio-based fertilizers in the Latvian economy in 2015, in million kg.

## 5. Conclusions

The paper represents the first attempt to close the gap in the data to measure the bioeconomy in Latvia. In some cases, it required arbitrary assumptions to be made to compile a BFM for Latvia and identify the data gaps for further research.

With available data, this approach can be applied to any country aiming to measure the size of its bioeconomy.

According to the results of the bio-based material flow monitor (BFM), we conclude that the bioeconomy includes more sectors than mentioned in previous reviews of the literature. This should be considered by Latvian policy makers when defining the bioeconomic strategy for the next period. There are significant differences when measuring the bioeconomy in terms of monetary values and physical volumes, which emphasizes the importance of high-added-value sectors in the bioeconomy.

There are some shortcomings of the use of BioSAM. Initial BioSAM supply and use tables are not balanced. The BioSAM has a higher disaggregation level than SUTs do; nevertheless, it does not cover all the sectors of national SUTs. For example, waste management is included in SUTs, but it is not included in BioSAM.

The definitions of BioSAM commodities should be elaborated to set correspondence with common nomenclature (CN) and Prodcom codes. Otherwise, the granularity may include a diverse set of commodities for a BioSAM commodity that hampers pricing.

To promote the credibility of the results, national statistics officers should be involved to validate the results of the approach used in this paper.

This study can be built upon by applying data regarding the total physical volumes of commodities that are mostly available by national statistics to BioSAM shares of commodities supplied/used by the different sectors. This might provide an opportunity for BioSAM tables in volumes that result in physical supply and use tables, to check the robustness of the results. The release of BioSAM 2020 data is expected in 2023, and it will provide opportunities to analyze more recent data and observe the dynamics.

**Author Contributions:** Conceptualization, V.T.; methodology, V.T.; validation, V.T. and S.Z.-R.; formal analysis, V.T. and S.Z.-R.; investigation, V.T.; resources, V.T.; data curation, V.T.; writing—original draft preparation, V.T. and S.Z.-R.; writing—review and editing, V.T. and S.Z.-R.; visualization, V.T. and S.Z.-R.; supervision, V.T.; project administration, S.Z.-R.; funding acquisition, S.Z.-R. All authors have read and agreed to the published version of the manuscript.

**Funding:** This research was funded by the BioMonitor project (http://biomonitor.eu), which received funding from the European Union's Horizon 2020 research and innovation programme under grant agreement No. 773297. The APC was funded by the BioMonitor project.

**Institutional Review Board Statement:** Not applicable.

**Informed Consent Statement:** Not applicable.

**Data Availability Statement:** The data presented in this study are available upon request from the corresponding author. The data are not publicly available due to ongoing research.

**Conflicts of Interest:** The authors declare no conflict of interest.

## Abbreviations

| | |
|---|---|
| BFM | bio-based material flow monitor |
| BI | balancing item |
| CN | common nomenclature |
| CPA | classification of products by activity |
| EAA | economic accounts for agriculture |
| EC | European Commission |
| EU | European Union |
| GVA | gross value added |
| IO | input–output |
| JRC | Joint Research Centre |
| MFA | material flow account |
| MFM | material flow monitor |
| MSUT | monetary supply-use table |
| NACE | statistical classification of economic activities |
| PSUT | physical supply and use table |
| SAM | social accounting matrix |
| SN | Statistics Netherlands |
| SUT | supply and use table |

## Appendix A

**Table A1.** Classification of sectors and commodities in standard SAMs [13].

| Sectors and Commodities | |
|---|---|
| Agriculture | Machinery and equipment nec |
| Forestry | Manufactures nec |
| Fishing | Electricity and gas |
| Coal | Water |
| Food industry | Construction |
| Textiles, wearing, leather | Trade |
| Wood products | Transport nec |
| Paper products, publishing | Water transport |
| Petroleum, coal | Air transport |
| Chemical, rubber, plastic products | Communication |
| Mineral products nec | Financial services nec |
| Metals | Insurance |
| Metal products | Business services nec |
| Motor vehicles and parts | Recreational and other services |
| Transport equipment nec | Public administration, defense, education, health |
| Electronic equipment | Dwellings |

**Table A2.** Sectors and commodities of BioSAM accounts [13].

| Sectors and Commodities | |
|---|---|
| Paddy rice | Processed sugar |
| Wheat | Prepared animal feeds |
| Barley | Other food products |
| Grain maize | Wine |
| Other cereals | Other beverages and tobacco |
| Tomatoes | Textiles, apparel, and leather |
| Other vegetables | Wood products |
| Grapes | Pellets |
| Fruits and nuts | Paper products, publishing |
| Rapeseeds | Petroleum, coal |
| Sunflower seed | Chemical, rubber, plastic products (non-bio-based) |
| Soya seed | Biogasoline |
| Olive for the oil industry | Biodiesel |
| Other seed for the oil industry | 2nd-generation biofuel–biochemical pathway fuels * |
| Sugar beet | 2nd-generation biofuel–thermal pathway fuels * |
| Fibre plants | Fertilizers |
| Potatoes | Biochemicals * |
| Live plants | Mineral products nec |
| Fodder crops | Metals |
| Tobacco | Metal products |
| Other crops | Motor vehicles and parts |
| Bovine cattle, live | Transport equipment nec |
| Sheep, goats, horses, asses (live) | Electronic equipment |
| Swine, live | Machinery and equipment nec |
| Poultry, live | Manufactures nec |
| Other animals, live, and their products | Electricity and gas ** |
| Raw milk | Bioelectricity ** |
| Forestry * | Water |
| Plantations * | Construction |
| Fishing | Trade |
| Mining | Transport nec |
| Meat of bovine animals | Water transport |
| Meat of swine | Air transport |
| Meat of sheep, goats, and equines | Communication |
| Meat and edible offal of poultry | Financial services nec |
| Vegetable oils and fats | Insurance |
| Olive oil | Business services nec |
| Oil-cakes | Recreational and other services |
| Dairy products | Public administration, defense, education, health |
| Rice, milled or husked | Dwellings |

* Due to a lack of price information, the commodity *Plantations* was added to *Forestry*; the commodity *2nd generation biofuels* was added to *Biochemicals*. ** The commodities *Electricity and gas* were replaced with *Natural gas* (volumes from PEFA, with conversion rate 1 terajoule = 46.52 t), and *Bioelectricity* with *Biogas* (1 TJ = 20.00 T).

## Appendix B

**Table A3.** Bio-based shares of BioSAM commodities.

| Commodity | Bio-Based Share, % | Commodity | Bio-Based Share, % |
|---|---|---|---|
| Paddy rice | 100.00 | Meat of sheep, goats, and equines | 100.00 |
| Wheat | 100.00 | Meat and edible offal of poultry | 100.00 |
| Barley | 100.00 | Vegetable oils and fats | 100.00 |
| Maize | 100.00 | Olive oil | 100.00 |
| Other cereals | 100.00 | Dairy products | 100.00 |
| Tomatoes | 100.00 | Rice, milled or husked | 100.00 |
| Other vegetables | 100.00 | Processed sugar | 100.00 |
| Grapes | 100.00 | Prepared animal feeds | 100.00 |

**Table A3.** *Cont.*

| Commodity | Bio-Based Share, % | Commodity | Bio-Based Share, % |
|---|---|---|---|
| Fruits and nuts | 100.00 | Other food products | 100.00 |
| Rapeseeds | 100.00 | Wine | 100.00 |
| Sunflower seed | 100.00 | Other beverages and tobacco | 75.00 |
| Soya seed | 100.00 | Textiles, wearing apparel and leather | 80.00 |
| Olive for the oil industry | 100.00 | Wood products | 100.00 |
| Other seed for the oil industry | 100.00 | Pellets | 100.00 |
| Sugar beet | 100.00 | Paper products, publishing | 100.00 |
| Fiber plants | 100.00 | Petroleum, coal | 0.00 |
| Potatoes | 100.00 | Chemical, rubber, plastic products (non-bio-based) | 0.00 |
| Live plants | 100.00 | Biogasoline | 100.00 |
| Fodder crops | 100.00 | Biodiesel | 85.00 |
| Tobacco | 100.00 | Fertilizers | 0.13 |
| Other crops | 100.00 | Biochemicals | 100.00 |
| Bovine cattle, live | 100.00 | Mineral products nec | 0.00 |
| Sheep, goats, horses, asses, . . . (live) | 100.00 | Metals | 0.00 |
| Swine, live | 100.00 | Metal products | 0.00 |
| Poultry, live | 100.00 | Motor vehicles and parts | 0.00 |
| Other animals, live, and their products | 100.00 | Transport equipment nec | 0.00 |
| Raw milk | 100.00 | Electronic equipment | 0.00 |
| Forestry | 100.00 | Machinery and equipment nec | 0.00 |
| Fishing | 100.00 | Manufactures nec | 0.00 |
| Mining | 0.00 | Natural gas | 0.00 |
| Meat of bovine animals | 100.00 | Biogas | 100.00 |
| Meat of swine | 100.00 | | |

## Appendix C

**Table A4.** NACE * sectors that are not included in BioSAM.

| NACE Code and Sector |
|---|
| C18—Printing and reproduction of recorded media |
| C21—Manufacture of basic pharmaceutical products and pharmaceutical preparations |
| C26—Manufacture of computer, electronic and optical products |
| C33—Repair and installation of machinery and equipment |
| H52—Warehousing and support activities for transportation |
| H53—Postal and courier activities |
| I—Accommodation and food service activities |
| K66—Activities auxiliary to financial services and insurance activities |
| L—Real estate activities |
| M71—Architectural and engineering activities; technical testing and analysis |
| M72—Scientific research and development |
| M73—Advertising and market research |
| M74–M75—Other professional, scientific and technical activities; veterinary activities |
| N77—Rental and leasing activities |
| N78—Employment activities |
| N79—Travel agency, tour operator, and other reservation service-related activities |
| N80–N82—Security and investigation, service and landscape, office administrative, and support activities |
| Q87–Q88-Residential care activities and social work activities without accommodation |
| S94—Activities of membership organizations |
| S95—Repair of computers and personal and household goods |
| S96—Other personal service activities |
| T—Activities of households as employers; undifferentiated goods- and services-producing activities of households for their own use |
| U—Activities of extraterritorial organizations and bodies |

* NACE—Statistical classification of economic activities.

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
