# Peer review of "Closing Data Gaps to Measure the Bioeconomy in the EU"

_2673-8783, doi:10.3390/biomass3020008_

Round 1

Reviewer 1 Report

"The aim of the paper is to develop bio-based material flow monitor to measure the physical contribution of industries to the 13 bioeconomy" -- consider different wording. You did not develop a monitor (i.e., a measurement device), you took an approach to estimated the flow of biobased materials in Latvia.

The title suggests something far broader than the study. Ultimately, this study is less about data issues than it is about (as the discussion says very concisely): "The paper is the first attempt to close the gap of missing data to measure the bioeconomy in Latvia that, in some cases, required to make arbitrary assumptions to compile BFM for Latvia and identify the data gaps for further research." The title should reflect that.

Author Response

I appreciate the suggestion made by the reviewer to improve the paper.

As suggested by the reviewer, we changed the title of the paper to ‘Closing data gaps to measure the bioeconomy in the EU’.

Reviewer 2 Report

The manuscript titled "Facing data issues in measuring the bioeconomy in EU" is the first attempt to close the gap of missing data to measure the bioeconomy in Latvia. The manuscript requires major changes before publication. Here are some comments/suggestions: 

1. Title: The title should be changed to be more specific about the information presented in the manuscript since the authors are more focused on presenting the bioeconomy in Latvia. Then, the title should be changed accordingly. 

2. Introduction. This section provides good information about the bioeconomy data issues and the statement of the novelty of the manuscript. Nevertheless, the bioeconomy definition given by the EU should be explained deeply. 

3. Measuring bioeconomy in EU and Latvia. This section should show the impact of different methods to measure the bioeconomy in the EU. Moreover, the authors should focus better on contextualizing Latvia in the manuscript. Please, add a reference in line 79. In addition, change Figure 1 to see all the sector names (i.e., avoid ellipsis). The authors can include how national and international contexts can affect biomass trade (and then bioeconomy). 

4. Materials and methods. This section could be explained deeply since the authors are describing they made the calculations. Assumptions should be presented in this section. Can the methodology for estimating a bio-based material flow monitor (BFM) be applied to other countries? If yes, presenting a step-by-step guide or methodology could be a good option. This can give the journal readers a tool to estimate a bio-based material flow monitor for other countries.

5. Resutls, Discussion, and Conclusions. This section must be improved since there is no broad discussion/comparison of the results presented. 

6. Conclusions should be another section at the end of the manuscript. Moreover, some lines dedicated to improving the proposed methodology can be added (perspectives and future work). 

7. English should be revised since there are wording sentences and the use of personal pronouns in the text (change to passive voice).

Author Response

We appreciate the suggestions made by the reviewer to improve the paper.

Response to comments:

  1. We changed the title of the paper to ‘Closing data gaps to measure the bioeconomy in the EU’.
  2. We extended the introduction by including more details on the bioeconomy definition.

  3. As we mentioned in the paper, there are no methods to compare, as the aggregation level of available data is high and the data are not available for biomass estimation in volume. Unless the corresponding author has developed an approach in the paper that is under revision, and still it is in monetary value, not volumes. Similar research was done by Cingiz et al. and Lazorcakova et al. as mentioned in the paper, with monetary outcomes.

    The reference is added to the sentence on line 79.  The Figure 1 is fixed. We introduced dynamics in Table 1 for a better description of the bioeconomy in Latvia.

    Assessment of the impact of international trade of biomass, in light of recent geopolitical issues in the region, is impossible due to the lack of more recent data. We look forward to working on it in the future as this is highly important to be assessed. 
  4. The Figure 2 is introduced in this section to show a step-by-step procedure on how to apply this approach to any country BioSAM data set for bio-based material flow monitoring.

  5. We combined Results and discussions as one section of the paper.
  6. Section Conclusions deals with conclusions, shortcomings, and future work.

  7. The use of English was improved. Sentences with personal pronouns are changed to passive voice.

Reviewer 3 Report

The work proposes a methodology to measure the contribution of bio-based material flows to the bioeconomy. The described procedure was applied to the case of Latvia. The article is well written, the results are well supported by the data, but overall it is not easy to read, probably also due to the high number of abbreviations. Probably a list of symbols and abbreviation can help the reader.

The caption of the figures and tables also need to be improved. As general rule they should be self-explanatory

In figure 1 there are some sectors which are truncated and are not readable

Please insert a white line between caption (of figures 3, 4, 6 and 7) and text.

Author Response

I appreciate the suggestions made by the reviewer to improve the paper.

We introduced Appendix C of abbreviations.

The captions of tables and figures are improved.

Figure 1 is fixed. White lines between captions and text are inserted.

Round 2

Reviewer 2 Report

The manuscript can be published in the current version.